# Robustness Analysis of the Estimators for the Nonlinear System Identification

**DOI:** 10.3390/e22080834

**Published:** 2020-07-30

**Authors:** Wiktor Jakowluk, Karol Godlewski

**Affiliations:** Faculty of Computer Science, Bialystok University of Technology, Wiejska 45A, 15-351 Bialystok, Poland; godlewski215@gmail.com

**Keywords:** robust estimation, system identification, model fitting, optimal control

## Abstract

The main objective of the system identification is to deliver maximum information about the system dynamics, while still ensuring an acceptable cost of the identification experiment. The focus of such an idea is to design an appropriate experiment so that the departure from normal working conditions during the identification task is minimized. In this paper, the adaptive filtering (AF) scheme for plant model parameter estimation is proposed. The experimental results are obtained using the nonlinear interacting water tanks system. The adaptive filtering is expressed by minimizing the error between the system and the plant model outputs subject to the white noise signal affecting system output. This measurement error is quantified by the comparison of Minimum Error Entropy Renyi (MEER), Least Entropy Like (LEL), Least Squares (LS), and Least Absolute Deviation (LAD) estimators, respectively. The plant model parameters were obtained using sequential quadratic programming (SQP) algorithm. The robustness tests for the double-tank water system parameter estimators are performed using the ellipsoidal confidence regions. The effectiveness analysis for the above-mentioned estimators relies on the total number of iterations and the number of function evaluation comparisons. The main contribution of this paper is the evaluation of different estimation methods for the nonlinear system identification using various excitation signals. The proposed empirical study is illustrated by the numerical examples, and the simulation results are discussed.

## 1. Introduction

System identification is generally performed by perturbing processes under operation and using the measurement data to build the system model. The main objective of the system identification is to excite the system under consideration using an applicable input and construct the plant model with maximum accuracy [1,2]. The identification experiment can be performed in both closed and open-loop systems. The inappropriate experimental conditions can lead to the performance loss of the loop to be controlled. It has been noticed that about 80% of the developed control loops does not ensure the desired performance assessment [3].

The estimators of various objective functions can be compared using mean square errors. In many applications, the main objective is to find an unbiased estimator that should have minimum variance. The main focus of such an experiment is to design the optimal input signal that maximizes selected measures based on the Fisher information matrix (FIM) [4]. The motivation for this approach is the Cramer–Rao definition where the covariance matrix of the parameter estimates converges to the inverse of the FIM.

In recent years, a significant effort has been made to develop identification methods for robust control [5]. The robust system identification consists of the real model parameter estimation and the bounds imposed on the model uncertainty set [6]. Applying the worst-case identification experiment to model parameter estimation, the error bounds are established in the form of the noise affecting the model of the system [7]. However, the above-presented methods do not guarantee the acceptable control loop performance. The reason for this inconvenience is the fact that the experimental conditions should be chosen in such a way that the identified model is precise.

This problem can be solved by the least-costly, application-oriented, and plant-friendly identification experiment for control. The main goal of the least-costly experiment is to construct the uncertainty set that is possibly small and provides the best control performance. Such a kind of experiment requires the identification cost to be connected with the input design through the objective function [8]. The application-oriented experiment design is quantified by first computing an optimal input spectrum, and then constructing a perturbation signal required by real working conditions [9]. The plant-friendly experiment design is comparable to the application-oriented input design technique. The objective of this experiment is to find a trade-off between minimal departure from real working conditions and the precision of the model parameters to be identified [10,11]. For this purpose, the concept of the performance degradation minimization instead of the variance minimization should be considered.

When the optimal input signal has been obtained, one can identify the system under control in a sequential manner. Then, the objective is to sequentially improve the system performance by more precise estimates of the plant model parameters. This issue can be solved by the model predictive control (MPC) and adaptive filtering (AF) techniques where the input signals are obtained to guarantee the optimal control performance [12,13]. The MPC and AF formulations have been widely used in a great number of industrial applications because of an increase in the speed of processors. The MPC scheme contains the discrete-time nominal model of the system, MPC optimizer, and the Kalman filter (KF) as a prediction model [14,15]. The MPC technique is based on the prediction of system succeeding outputs subject to the future input signals. The advantage of MPC is the ability to simply operate constraints on control signals and state variables. The adaptive filtering method can be formulated as minimizing the error between the system output and the adaptive filter output with unknown parameters. The true system output can be perturbed using Gaussian or no-Gaussian noise models. The main objective of such an experiment is to extract the plant model with maximum accuracy.

The often-used adaptive methods such as Least Squares (LS) and Least Absolute Deviation (LAD) may operate incorrectly in terms of mean square error and convergence speed. The LS estimator is computationally simple and mathematically tractable but is often criticized for its noticeable lack of robustness. In effect, one single outlier can have a great impact on the estimate. To receive a more robust regression estimator, the LAD criterion has been introduced [16,17]. The LAD method is robust because it is resistant to the data outliers, but it can lead to multiple unstable solutions. The different estimators where the penalty function is minimized subject to the overall distribution of residuals are Least Median of Squares (LMS), Least Trimmed Squares (LTS), and Reweighted Least Squares (RLS) [18]. The Maximum Likelihood (ML), Minimum Entropy (ME), and Generalized Maximum Entropy (GME) methods for robust parameter estimation, which guarantee robustness subject to regression models, are proposed in [19]. Another prediction error estimation method called the Least Entropy-Like (LEL) estimator is described in [20,21]. This method is based on properly established penalty function and is developed based on the Gibbs entropy definition. The main goal of this algorithm is to examine the global dispersion measure of the residuals fit. The LEL estimator is robust to outlying data because the cost function is minimized subject to relative squared error variability. The Minimum Error Entropy Renyi (MEER) is adopted in many applications as a generalization of Shannon entropy [22]. The Renyi entropy for several values of *α* takes on different forms. When *α* equals zero, the Renyi entropy becomes the Hartley entropy. For *α* equal to one, the Renyi entropy becomes standard Shannon entropy. In practice, Renyi entropy of order two is just called collision entropy and is often used in qualitative tests for random number generators [23,24]. Finally, when *α* tends to infinity, the Renyi entropy is increasingly determined by the events of the highest probability.

The focus of this paper is to provide the robustness analysis of the LS, LAD, LEL, and MEER methods for the nonlinear interacting water tanks system parameter estimation. The parameters of the double water tank model are selected arbitrarily to ensure the gravitational flow of water in the tanks. Many real-world systems indicate a quasi-linear or nonlinear manner during normal operation and exhibit a hard saturation effect for high peaks of the input signal. The standard examples of such systems are the water tank processes [25,26]. Nonlinear state-space modeling is a promising, and at the same time challenging, category of techniques [25]. In this paper, the adaptive filtering method for the parameter estimation of the nonlinear state-space model, the cascaded water tanks, is verified [24]. Suitability and performance tests to capture the dynamical behavior of the water tank process are performed. The effectiveness evaluation for the estimators under consideration relies on the total number of iterations and the number of function evaluation comparisons. There are several works related to water tank system identification [27,28,29], but such research comparing the different estimators for the nonlinear system parameter identification has not been performed.

This paper is structured as follows. In Section 2, the problem statement of the unknown system parameter estimation is derived. In Section 3, the nonlinear dynamic model of the double water tank process is presented. The LS, LAD, LEL, and MEER estimators are discussed in detail in Section 4. The numerical results and discussion for the second-order dynamic model parameter identification are presented in Section 5. In this section, the simulation effects for chosen estimation methods, different excitation signals, and various system initial conditions are studied. The identification duration based on the total number of iterations and function evaluations is verified. Finally, some concluding comments are made in Section 6.

## 2. Problem Statement

Consider the nonlinear system formulated as follows:(1)y(t)=f(u,t,θ,ε),
where *y(t)*, *u(t)*, *ε(t),* and *θ* are: output, input, noise, and parameter vector of the true system, respectively. System identification is the process of constructing a mathematical model of the system from experimental data and a priori knowledge about the system. The accuracy of the model parameter estimates is dependent on the input signal selection [30,31]. The scheme of the system parameter identification is shown in Figure 1 we perturbed the system′s input using *u*(*t*), and we observed data on its output *y*(*t*).

Referring to Figure 1, the algorithm can be defined as the error *e*(*t*) minimization between the true system response and the adaptive filter output [24]. The output of the system is affected by the white noise signal with zero mean and finite variance values. The system with nominal parameter values and the adaptive filter with unknown parameters were perturbed using selected continuous-time input signals. The system was assumed to be at the initial states and control signal conditions. The unknown parameters of the adaptive model were computed, in an iterative way, for different initial state conditions and various white noise variance values. The adaptive model parameters were obtained using the interior point method of constrained optimization. The goal of such an experiment was to estimate adaptive model parameter values which should be comparable to the real values of the system parameters. The constrained objective functions built based on LS, LAD, LEL, and MEER estimators were minimized to obtain the most precise parameters of the adaptive model. The accuracy of the above estimators was verified using absolute, relative, and mean square errors, respectively. The graphical representation of the adaptive model parameter estimates for different input signals are shown by ellipsoidal confidence regions. The performance assessment of the estimation methods was performed comparing the total number of iterations and the number of function evaluations. The results of this identification experiment can answer the estimator selection for instance for MPC implementation.

## 3. The Nonlinear Interacting Water Tanks Process

In this section, the second-order nonlinear dynamic system is presented in Figure 2. The gravitational interacting tanks system consists of two interconnected cylindrical water tanks [29]. The model of an individual cylindrical tank is described by the volumetric flow *Q_in_*(*t*) into the upper tank to the water outflow *Q_out_*(*t*) through the valve of the lower tank. The balance of the water flow in each tank can be defined as:(2)$$A\frac{dh(t)}{dt}=Q_{in}(t)-Q_{out}(t),$$
where: *A* is the cross-sectional area of each tank and *h*(*t*) is the water level in the tank.

It has been established that the outlet leak has an ideal sharp-edged orifice. The water outflow of the connected tanks can be described by the Torricelli′s law, given by the following equation:(3)$$Q_{out}(t)=a\cdot \sqrt{2gh(t)}\;.$$

In the above equation, *a* is the cross-sectional area of the orifice (in some approaches, it is also multiplied by the constant *C_d_*, called the discharge coefficient of the valve, which takes into account all fluid characteristics, losses, and irregularities in the system) and *g* denotes the gravitational constant (9.8 m/s^2^).

Substituting Equation (2) and (3) and assuming that the output hole of the first tank is the input to the second one, the nonlinear model of the system is as follows:(4)$$\left\{ \begin{array}{ll} A_{1}\frac{dh_{1}(t)}{dt}=-a_{1}\cdot \sqrt{2gh_{1}(t)}+Q_{in}(t), & h_{1}\left(0\right)=h_{10},\\ A_{2}\frac{dh_{2}(t)}{dt}=a_{1}\cdot \sqrt{2gh_{1}(t)}-a_{2}\cdot \sqrt{2gh_{2}(t)}, & h_{2}\left(0\right)=h_{20}. \end{array}\right.$$

The index *n* = 1, 2 signifies the number of the tank in the tank cascade. The Equation (4) represents the mathematical model and characterizes the behavior of a second-order nonlinear dynamic system.

The differential Equation (4), after the following substitution, can be written in the standard form of the state-space equations: *Q_in_*(*t*) = *u*(*t*), *x*_1_(*t*) = *h*_1_(*t*), *x*_2_(*t*) = *h*_2_(*t*), *y*(*t*) = *h*_1_(*t*).
(5)$$\begin{array}{l} \left\{ \begin{array}{ll} \frac{dx_{1}(t)}{dt}=-\frac{a_{1}}{A_{1}}\cdot \sqrt{2gx_{1}(t)}+\frac{1}{A_{1}}u(t), & x_{1}\left(0\right)=h_{10},\\ \frac{dx_{2}(t)}{dt}=\frac{a_{1}}{A_{2}}\cdot \sqrt{2gx_{1}(t)}-\frac{a_{2}}{A_{2}}\cdot \sqrt{2gx_{2}(t)}, & x_{2}\left(0\right)=h_{20}, \end{array}\right.\\ y(t)=x_{1}(t), \end{array}$$
where: *x*_1_ = *x*_1_(*t, a*_1_), *x*_2_ = *x*_2_(*t, a*_1_, *a*_2_).

The steady-state condition (*x*_1*std*_, *x*_2*std*_) in response to the constant input signal *u_std_* can be calculated from the set of algebraic equations:(6)$$\left\{ \begin{array}{l} 0=-\frac{a_{1}}{A_{1}}\cdot \sqrt{2gx_{1std}}+\frac{1}{A_{1}}u_{std},\\ 0=\frac{a_{1}}{A_{2}}\cdot \sqrt{2gx_{1std}}-\frac{a_{2}}{A_{2}}\cdot \sqrt{2gx_{2std}}, \end{array}\right.$$
then we obtain:(7)$$x_{1std}=\frac{1}{2g}{\left(\frac{u_{std}}{a_{1}}\right)}^{2},\;x_{2std}={\left(\frac{a_{1}}{a_{2}}\right)}^{2}\cdot x_{1std}=\frac{1}{2g}{\left(\frac{u_{std}}{a_{2}}\right)}^{2}.$$

If we assume that the cross-sectional areas of the tanks can be exactly determined, the parameters to be estimated are the cross-sectional areas of the orifices: *a*_1_ = *a, a*_2_ = *b*. Then, the state-space equations of the double tank system can be expressed in the following form:(8)$$\left\{ \begin{array}{ll} {\dot{x}}_{1}=-\frac{a}{A_{1}}\cdot \sqrt{2gx_{1}}+\frac{1}{A_{1}}u, & x_{1}\left(0\right)=h_{10},\\ {\dot{x}}_{2}=\frac{a}{A_{2}}\cdot \sqrt{2gx_{1}}-\frac{b}{A_{2}}\cdot \sqrt{2gx_{2}}, & x_{2}\left(0\right)=h_{20}, \end{array}\right.$$
where: *x*_1_ = *x*_1_(*t, a*), *x*_2_ = *x*_2_(*t, a, b*) and the state coordinates (the water levels in the tanks *h_n_*(*t*), *n* = 1, 2) have the natural physical constraints:(9)$$h_{n,\;max}\geq x_{n}(t)\geq 0,\;n=1,\;2.$$

The nonlinear double tank model of the true system was executed using a Matlab–Simulink environment [27]. The Simulink block diagram of the system under consideration is shown in Figure 3.

During double tank model parameter estimation, the difference between the system and the adaptive model outputs is minimized. According to Equation (5), the state variable *x*_1_ is equal to *h*_1_ (i.e., the water level in the first tank). Therefore, the difference between system and adaptive model water levels of the first tank should be minimized. Consequently, the marks 1 and 2 in Figure 3 indicate the integrator and the saturation block, respectively. The integrator block outputs the value of the integral of the water level concerning time, and the saturation block produces an output signal which is bounded to the upper and lower limits of the physical values. The parameters of the model are selected arbitrarily to ensure the gravitational flow of water through the tanks. The physical constraints and the model parameters of the water tank process are displayed in Table 1.

The chosen methods of estimation of the adaptive model parameters will be presented in the next section of this paper.

## 4. Robust Parameter Identification

The system identification for control is the process of the plant model development where outlying data have a significant impact on model parameter estimation. The general and well-accepted definition of outlying data does not exist. In considerations that follow, it was assumed that the outlying data should be inconsistent with the rest of the set.

In this paper, the Least Squares (LS), Least Absolute Deviation (LAD), Least Entropy Like (LEL), and Minimum Error Entropy Renyi (MEER) methods were considered for adaptive filter parameter estimation. The system identification block diagram is shown in Figure 1. The main concept of the prediction error method (PEM) is based on the objective function minimization subject to prediction error residuals [1].

Suppose that the data set consists of the points (*x_i_, y_i_*) with *i* = 1, 2,…, *l*. We want to find a function *f* such that *f*(*x_i_*) ≈ *y_i_*. To solve the mentioned optimization problem the following system was considered:(10)$$\begin{array}{l} y_{i}=f\left(x_{1},x_{2},\ldots ,x_{l},\theta_{r}\right)+\varepsilon_{i},\\ i=1,2,\ldots ,l, \end{array}$$
where: *θ_r_* is the vector of estimated parameters, *y_i_* is the system output sequence, and *ε_i_* is the Gaussian white noise with finite variance. The PEM estimators are calculated using the residuals:(11)$$r_{i}=y_{i}-{\hat{y}}_{i},$$
where ${\hat{y}}_{i}$ are the identified outputs i.e., ${\hat{y}}_{i}=f\left(x_{1},x_{2},\ldots ,x_{l},\hat{\theta }\right).$ The primary and frequently used method for regression analysis is the Least Squares estimation. The LS objective function minimizes the sum of squared residuals of the fit (i.e., prediction error estimator) is:(12)$${\hat{\theta }}_{LS}=\mathrm{Arg}\underset{\theta }{\min}\sum_{i=1}^{N}r_{i}^{2}=\mathrm{Arg}\underset{\theta }{\min}\left(r^{T}r\right),$$
where *r* is the residual vector of the fit $r=f\left(r_{1},r_{2},\ldots ,r_{N}\right).$ The similar technique, which is robust for model parameter estimation is Least Absolute Deviations (LAD) [16].
(13)$${\hat{\theta }}_{LAD}=Arg\underset{\theta }{\min}\sum_{i=1}^{N}\left|r_{i}\right|.$$

Similar to the LS method, LAD minimizes an objective function which should closely estimates a data set. This technique minimizes the sum of absolute errors (SAE). The sum of the absolute values minimizes the residuals between points obtained by the function and corresponding data points. The LAD is robust because it is resistant to outliers in the data points and guarantees equal emphasis in all observations. In comparison with ordinary LS which, by squaring the corresponding data points, ensures the larger weight of the residuals (i.e., the outliers in which estimated values are far from actual observations). The disadvantage of this method is multiple solutions.

The Least Entropy-Like (LEL) method was inspired by the idea of Gibbs entropy [20]. The concept behind this method is to examine the global dispersion value of the residuals fit. The PEM built according to Equation (11) is given by:(14)$$S=\sum_{j=1}^{N}r_{j}^{2},$$

The relative squared residuals are defined as:(15)$$\mathrm{if}\;S\neq 0\;\mathrm{then}\;s_{i}=\frac{r_{i}^{2}}{\sum_{j=1}^{N}r_{j}^{2}}\;\mathrm{where}\;s_{i}\in \left[0,1\right],\sum_{i=1}^{N}s_{i}=1,$$

Regarding paper [20], the estimation cost *Φ* based on relative squared residuals is defined as:(16)$$\Phi =\left\{ \begin{array}{ll} 0 & \mathrm{for}\;S=0\\ -\frac{1}{\log N}\sum_{i=1}^{N}s_{i}\log s_{i} & \mathrm{for}\;S\neq 0 \end{array}\right..$$

According to the entropy-like function *Φ* as Equation (16), the LEL estimator is proposed:(17)$${\hat{\theta }}_{LEL}=\mathrm{Arg}\underset{\theta }{\min}\Phi ,$$

Entropy-like expression, such as Equation (16), acts on the unknown parameters *θ* through the predictive error residuals. The LEL criterion Equation (17) is robust concerning outliers because the cost function is minimized subject to relative squared error dispersion. It should be noted that the penalty function Equation (16) is nonlinear and may not lead to a unique minimum for *θ.* Considering the LEL properties, one should first verify the LS quality fit. If the LS fit is perfect, there is no reason to use any other estimator. Finally, the LEL estimator should be executed numerically from initial conditions around the real parameter values.

The Minimum Error Entropy Renyi (MEER) concept can be specified as a generalization of the Shannon, Hartley, the collision, and the min-entropy [23]. The Renyi entropy expression is parameterized by a parameter *α*, which when allowed to approach unity, reverts to the well-known concept of Shannon. The MEER estimator for alpha in the range of 0 < *α* < ∞ is defined as follows:(18)$${\hat{\theta }}_{MEER}=\mathrm{Arg}\underset{\theta }{\min}\frac{1}{1-\alpha }\log\sum_{i=1}^{N}p_{i}^{\alpha },$$
where *α* is the Renyi entropy order and *p_i_* is normal probability density function with the mean value of zero and standard deviation of one, evaluated at the values in *r_i_* according to Equation (11). The probability distribution was computed using Matlab function *normpdf*. When *α* assumes the value of zero, the Renyi entropy becomes the Hartley entropy. For *α* approaching unity, the Renyi entropy reverts to standard Shannon entropy. In practice, Renyi entropy of order two is just called collision entropy. Finally, when *α* tends to infinity, the Renyi entropy is increasingly determined by the events of the highest probability. The parameter *α* of order two offers a large reduction in the computational effort.

The next section presents the results of the robustness tests for the double-tank water system parameter estimators using the ellipsoidal confidence regions. The effectiveness analysis for the above-mentioned estimators relies on the total number of iterations and the number of function evaluation comparisons.

## 5. Numerical Results and Discussion

To illustrate the advantages of the parameter identification process (Figure 1), using four different excitation signals *u*(*t*), we have selected a second-order nonlinear system—the cascade of two tanks. In this section, we discuss the formulation of the optimization problem where the error *e*(*t*) between the true system response and the adaptive filter output is minimized. For the numerical solution of this optimal control problem, a Matlab–Simulink environment [32] was employed.

For the system parameter estimation purposes, various input signals were adopted. The optimal input was computed for arbitrary selected nominal values of the system parameters: *a* = 0.05, *b* = 0.05, and with the termination time set as *T*_f_ = 10 s, using Riots_95 toolbox [33]. The system was assumed to be at the initial states: *x*_1_(0) = 0.75, *x*_2_(0) = 0.50, and the initial value of the input signal as *u*(0) = 0.05. Successive signals were selected as follows: step-input signal from the steady-state value of 0.20, sinusoidal signal with the frequency of 0.25 Hz, and steady-state value of 0.2 and sinusoidal signal with a frequency of 2.50 Hz and the same steady-state value.

The structure of the parameter identification process is shown in Figure 1, with the measurement noise of variance from the interval 0.0 ≤ *σ*^2^ ≤ 0.20. The plant model corresponds to the theoretical representation of the system Equation (8), which depends on a vector of unknown parameters ***θ*** = [*a*,*b*]^T^ to be estimated. The equations describing the system dynamics were integrated using the fixed step-size fourth-order Runge–Kutta method with grid intervals of 0.1 s. The one hundred and fifty runs were performed when the plant model starts from various initial state conditions, and the additional noise disturbing the system input has a different variance. The initial states of the system were chosen from the intervals 0.55 ≤ *x*_1_(0) ≤ 0.95 and 0.30 ≤ *x*_2_(0) ≤ 0.70. Four mentioned estimators and an interior-point method were used for the output error *e*(*t*) minimization.

Figure 4 and Figure 5 illustrate the number of iterations and function evaluations generated using the optimal input signal.

The above figures show the comparison of the number of iterations and function evaluations for different estimators when the system was perturbed using an optimal input signal. The main objective of this optimization task was to obtain the most accurate plant model parameters undertaking the minimum duration identification experiment. The results of the identification experiments in the form of mean iterations and mean function evaluations values for other input signals are summarized in Table 2 and Table 3.

It should be noted that the lowest mean values of iterations and function evaluations were obtained using optimal input excitation. Considering other input signals, the number of iterations and function evaluations increased significantly. The MEER algorithm yielded the smallest number of iterations and function evaluations. The following estimators are less robust and can be sorted in the order LS, LEL, and LAD. However, the advantage of LEL and LAD estimators is that these estimators find the optimal solution before reaching boundary values. Analyzing waveforms shown in Figure 4 and Figure 5, one can notice which estimator achieved optimal values in a minimal time lag.

Figure 6 illustrates the results of the identification experiments, i.e., the optimal values of parameters estimated as a solution of optimization tasks for each run. The ellipsoidal confidence regions were obtained for the LS, LAD, MEER, and LEL estimators using an optimal excitation signal. Analysis of the confidence regions of the nonlinear double tank model parameter estimates confirms some regularities. The optimal input signal yielded the minimal volume of the ellipsoidal confidence region for the MEER estimator. The cluster occupied by the model parameter values, computed using the LAD method, slightly increased its size for the same initial conditions and noise variance as in the previous experiment. The uncertainty regions obtained for the optimal input signal using LS and LEL estimators are not recommended in the real-life identification experiments.

For comparison, the ellipsoidal confidence regions obtained using the same estimators and step input signal are shown in Figure 7. The identification experiments were performed, when the system started from different initial conditions and the output of the system was disturbed by the white noise signal with different variance values. It can be noticed that for the nonoptimal step excitation signal, the clusters occupied by the model parameters estimated using the LAD and MEER methods increased their sizes. The plant model parameters estimated using sinusoidal perturbation signals were characterized by significant dispersion from nominal parameter values. It seems to be reasonable because sinusoidal input signals are generally used for frequency-domain system identification purposes.

Table 4 presents the percentage mean relative errors of the model parameter estimates for various perturbation signals and different estimation methods. Comparing the values of the mean relative errors shown in Table 4, it can be noticed that the LAD estimator occurred most precisely. However, considering the number of iterations and function evaluations, the MEER method has some advantages over the LAD algorithm. This is because the MEER duration of computation is shorter, and the identified parameters are practically similar. When the frequency of the sinusoidal excitation signal increases, the value of the percentage mean relative error also increases.

The LS method adapts to the most outlying data points from the average that can introduce the largest error value. If there exists a single disturbing outlier in the data very distant from the rest of the data points, it will attract a trend line to itself. The primary goal of the LEL method Equation (17) is to make the most of residuals striving to zero or to cause the relative squared residuals equally distributed concerning optimization criteria. The LEL method is a robust subject to outliers because the goal function to be minimized is a measure of the relative squared error variability. Finally, it can be noticed that there is no warranty for relative squared residual expression to have a unique solution subject to the parameters. So, the minimization should be executed carefully with particular attention to the initial values of the parameters.

Summarizing the simulation results, it should be noted that the LS estimator indicated maximum absolute error value equals to 0.497. The average value of the number of iterations was 3600 and the average function evaluation value was 13,500. The minimal absolute error value was observed for the LAD estimator equal to 0.232. The LAD method was worse in terms of the average number of iterations—5000 and the average function evaluations value—25,000. Interesting results were obtained for the MEER estimator where the maximum absolute error value was equal to 0.349. The corresponding amount of the average number of iterations was 2450, and the average function evaluation value was 9000. Surprising effects were received using the LEL estimator where the maximum absolute error value was about 0.475, the average number of iterations and the average function evaluation values were 3800 and 14,000, respectively. The reason for this inaccuracy is probably the lack of data outliers during the nonlinear double tank system parameter identification. The absence of the data outliers is related to the dynamic system initial condition selection.

To demonstrate that the results are structural and are not a function of the particular model parameters, the numerical experiments were repeated for additional initial parameter values *a* = *b* = 0.03 and *a* = *b* = 0.07, respectively. To report the results of numerical experiments, bar graphs were used. To simplify the analysis of the number of iterations, the number of function evaluations and the percentage relative errors the average values of estimated indicators, for four different input signals, were taken into consideration.

Figure 8, Figure 9 and Figure 10 contain data for the initial parameter values *a = b =* 0.05 presented in Table 2, Table 3 and Table 4. The simulation experiments for the additional initial parameter values of the nonlinear double tank system were performed under the same experimental conditions. It should be noted that despite the use of the average value of indicators for various input signals (i.e., sinusoidal signals which are not plant-friendly in the time domain identification), the results of the parameter estimation are similar. The least percentage mean relative errors (Figure 10) occurred for LAD and MEER estimators, respectively. The worst results regarding the number of iterations and the number of function evaluations were obtained for LAD and LEL estimators. Figure 10 shows some regularities: when the cross-sectional areas of the orifices increase their size, the percentage mean relative error decreases its value. Figure 11, Figure 12 and Figure 13 show the standard deviations of the mean values of the number of iterations, the mean values of the number of function evaluations, and the percentage mean relative error obtained as the average value of indicators for four different input signals using the LS, LAD, MEER, and LEL estimators, respectively. The standard deviation is a number used to demonstrate how the estimation results for a group are spread out from the average value. The low standard deviation value indicates that most of the numbers are close to the average. The above figures show that the greatest standard deviations for the number of iterations and the number of function evaluations were obtained for the LAD estimator. It should be noted that the MEER estimator had the greatest standard deviation of the percentage mean relative error for the small sizes of cross-sectional areas of the plant orifices. This inconvenience is caused by the percentage mean relative error outliers (i.e., very low error values) obtained using optimal perturbation signal. The advantages of the MEER estimator are low values of the standard deviations of the number of iterations and the number of function evaluations shown in Figure 11 and Figure 12.

## 6. Conclusions

The goal of the dynamic system identification task is to develop the appropriate experiment so that the departure from the system nominal working conditions is minimized. In this paper, the adaptive filtering method for the nonlinear interacting liquid tanks system identification was proposed. Adaptive filtering was defined by minimizing an error between the system and the model outputs subject to the Gaussian noise disturbing system output. This paper was devoted to the robustness analysis of the LS, LAD, MEER, and LEL estimators for the dynamic system parameter identification. The accuracy of the estimated adaptive filter parameters, for four estimation methods, different excitation signals, and various initial conditions was verified using ellipsoidal confidence regions. Effectiveness analysis was performed by comparison of the total number of iterations and the number of function evaluations per iteration. The main purpose of such research was to determine the most precise and effective estimator (e.g., for MPC tasks).

The performed experiments indicate that the best parameter estimation results were obtained using the LAD and MEER methods, respectively. The other presented methods produce worse estimation results: the LS estimator subject to the white noise affecting the system output and the LEL estimator subject to the lack of data outliers. It has been shown that the sinusoidal excitation signals are not recommended for double water tank system identification in the time domain. The simulation experiments performed for the additional initial parameter values demonstrate that the results are structural for interacting water tank process identification. The numerical results for standard deviation estimation also confirmed the effectiveness of the MEER estimator for second-order, nonlinear system identification. The most accurate results were obtained using the optimal input and the standard step input signals, respectively. In conclusion, based on the experiments performed, it should be stated that the best accuracy of the estimated parameters was observed for the optimal input signal and the LAD estimator. However, comparing parameter estimation durations, the MEER estimator should be considered for real-world identification experiments.

## Figures and Tables

**Figure 1 entropy-22-00834-f001:**
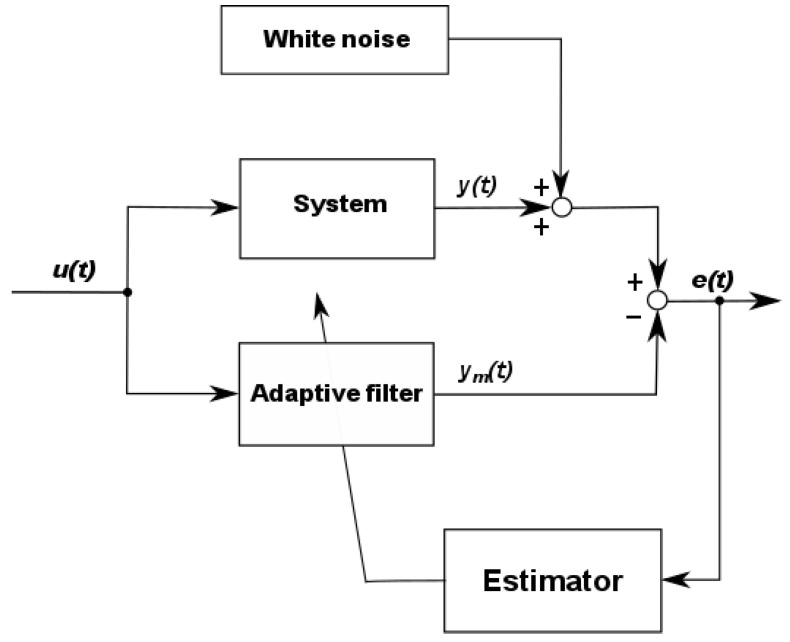
The system parameter estimation block scheme.

**Figure 2 entropy-22-00834-f002:**
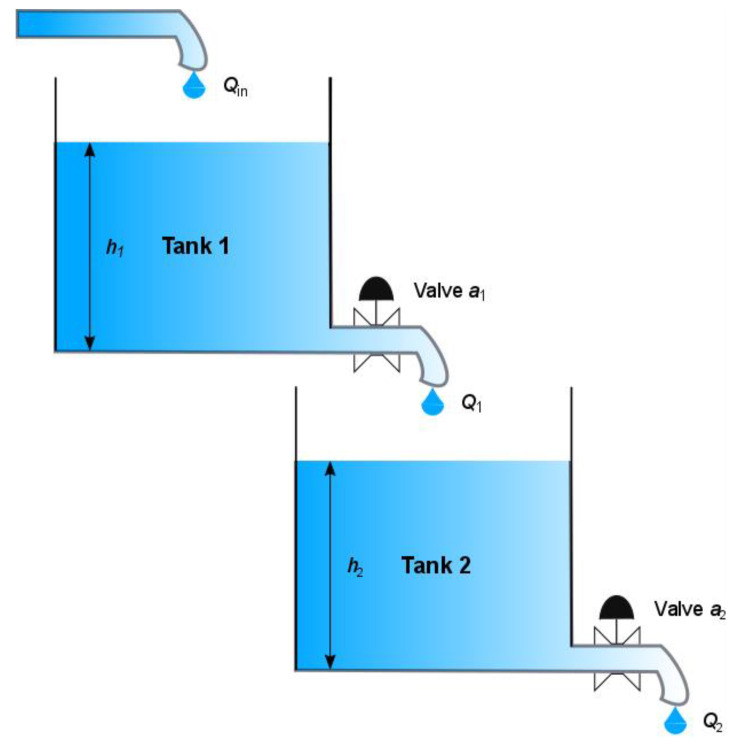
The double water tank process diagram.

**Figure 3 entropy-22-00834-f003:**
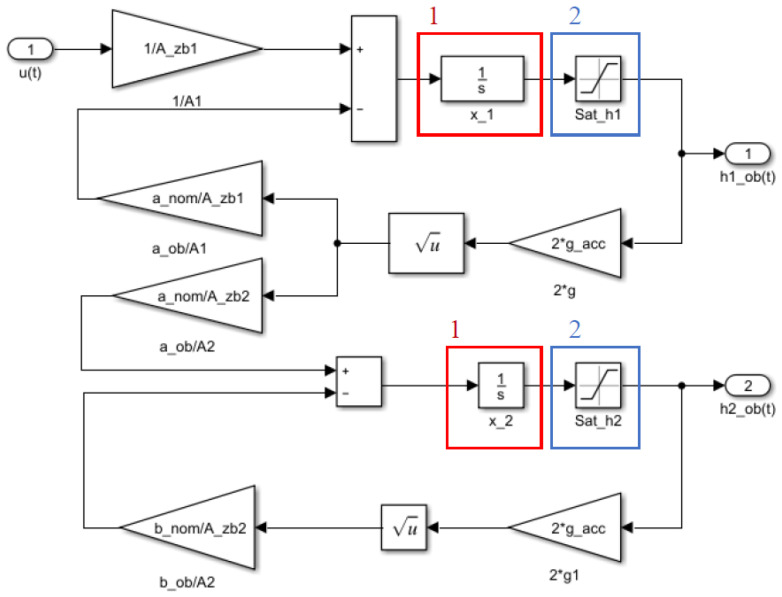
The Simulink block diagram of the interacting tanks system.

**Figure 4 entropy-22-00834-f004:**
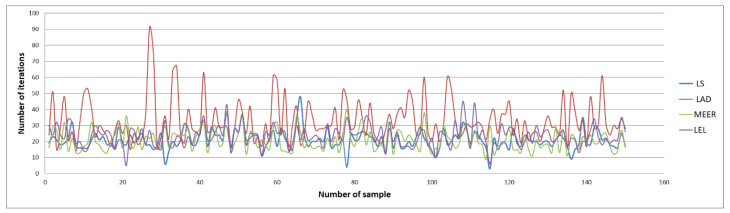
The comparison of the number of iterations obtained using the optimal input signal.

**Figure 5 entropy-22-00834-f005:**
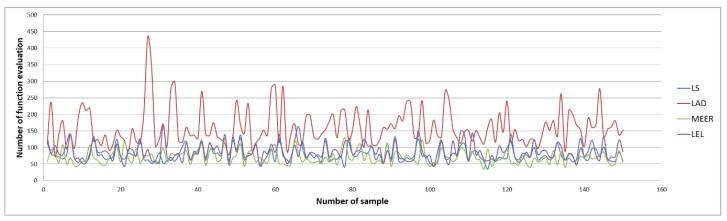
The comparison of the number of function evaluations obtained using the optimal input signal.

**Figure 6 entropy-22-00834-f006:**
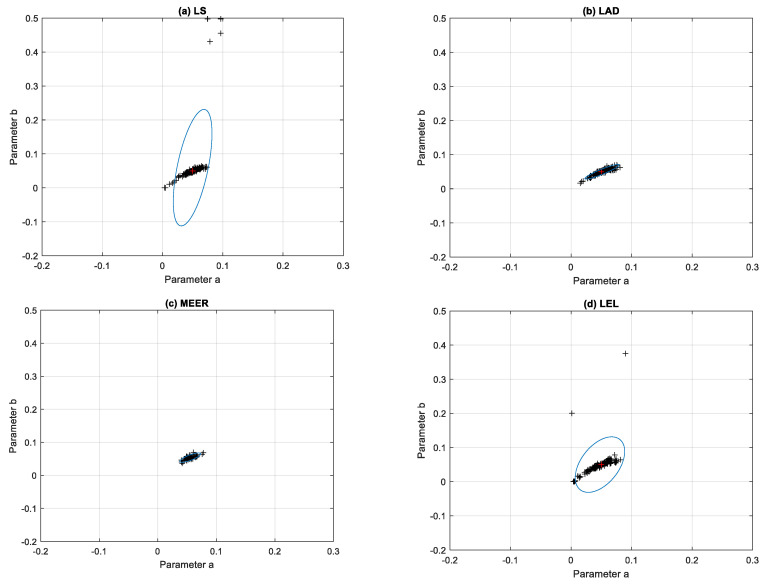
Ellipsoidal confidence regions obtained using: (**a**) Least Squares (LS), (**b**) Least Absolute Deviation (LAD), (**c**) Minimum Error Entropy Renyi (MEER), and (**d**) Least Entropy Like (LEL) estimators for the optimal input signal based on the 150 runs.

**Figure 7 entropy-22-00834-f007:**
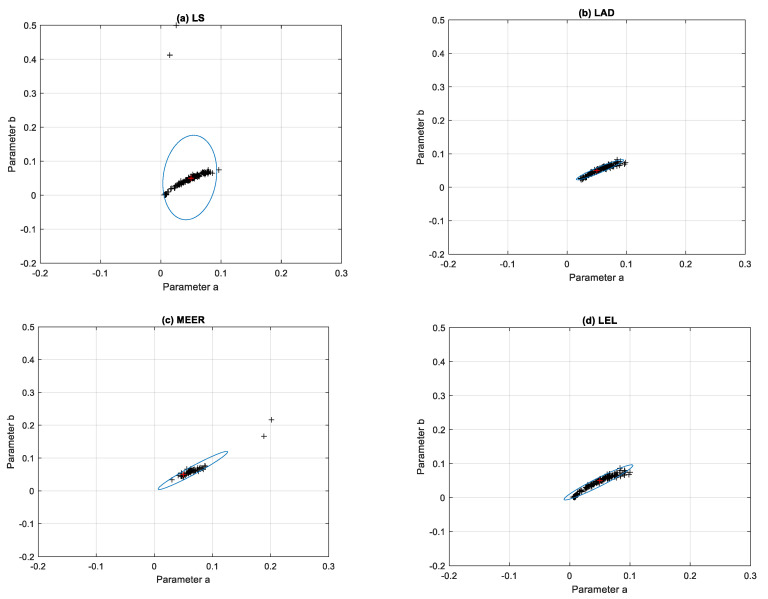
Ellipsoidal confidence regions obtained using: (**a**) Least Squares (LS), (**b**) Least Absolute Deviation (LAD), (**c**) Minimum Error Entropy Renyi (MEER), and (**d**) Least Entropy Like (LEL) estimators for the step input signal based on the 150 runs.

**Figure 8 entropy-22-00834-f008:**
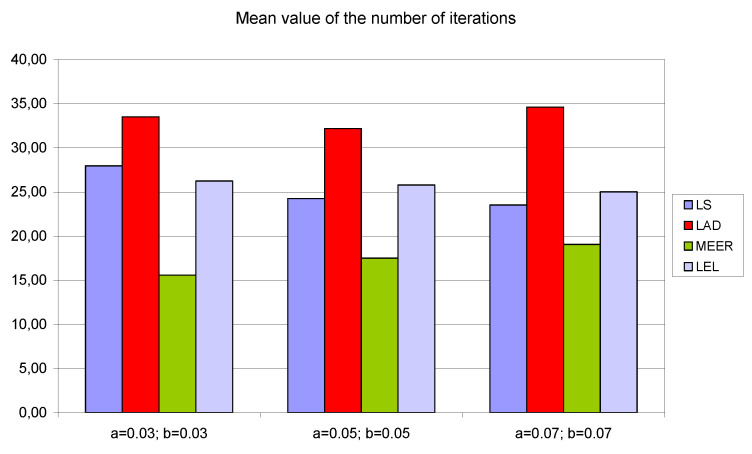
The mean values of the number of iterations obtained as the average value of indicators for four different input signals using LS, LAD, MEER, and LEL estimators.

**Figure 9 entropy-22-00834-f009:**
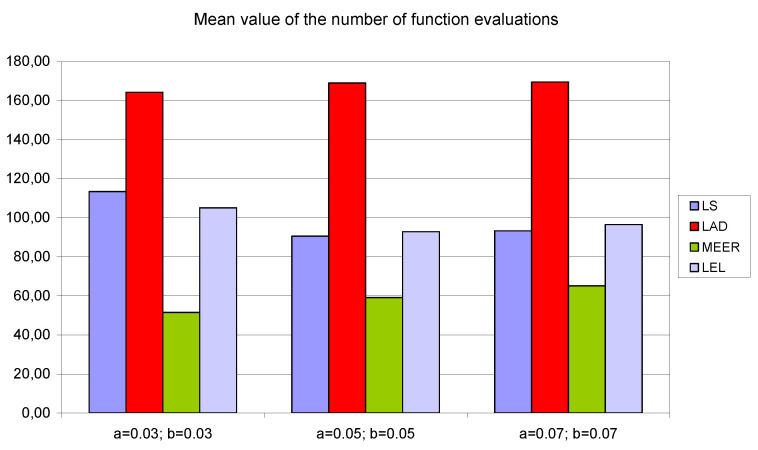
The mean values of the number of function evaluations obtained as the average value of indicators for four different input signals using LS, LAD, MEER, and LEL estimators.

**Figure 10 entropy-22-00834-f010:**
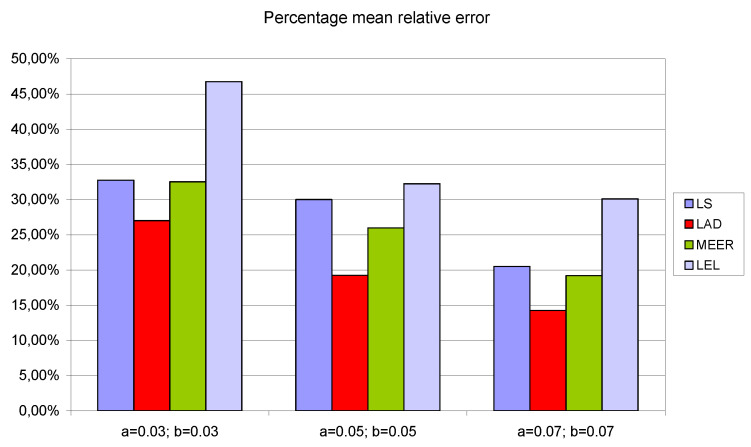
The percentage mean relative error obtained as the average value of indicators for four different input signals using LS, LAD, MEER, and LEL estimators.

**Figure 11 entropy-22-00834-f011:**
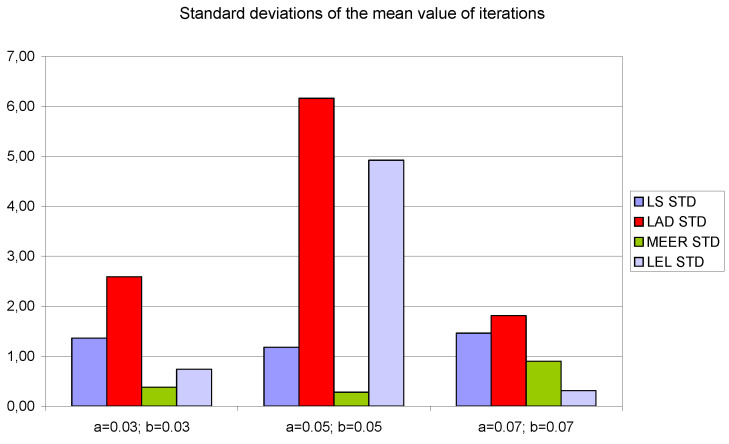
The standard deviations of the mean values of the number of iterations obtained as the average value of indicators for four different input signals using LS, LAD, MEER, and LEL estimators.

**Figure 12 entropy-22-00834-f012:**
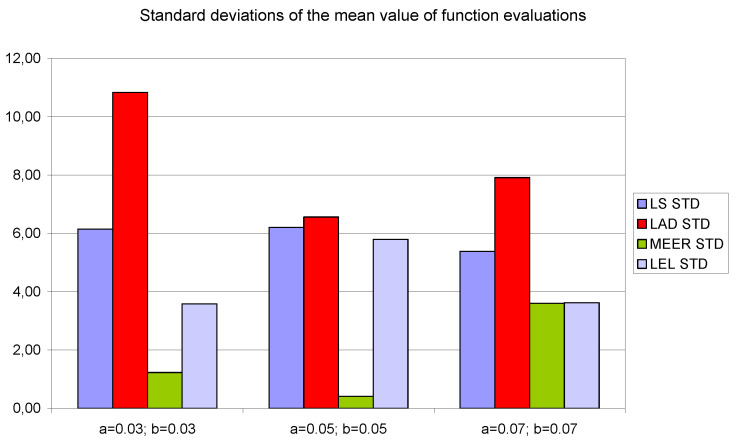
The standard deviations of the mean values of the number of function evaluations obtained as the average value of indicators for four different input signals using LS, LAD, MEER, and LEL estimators.

**Figure 13 entropy-22-00834-f013:**
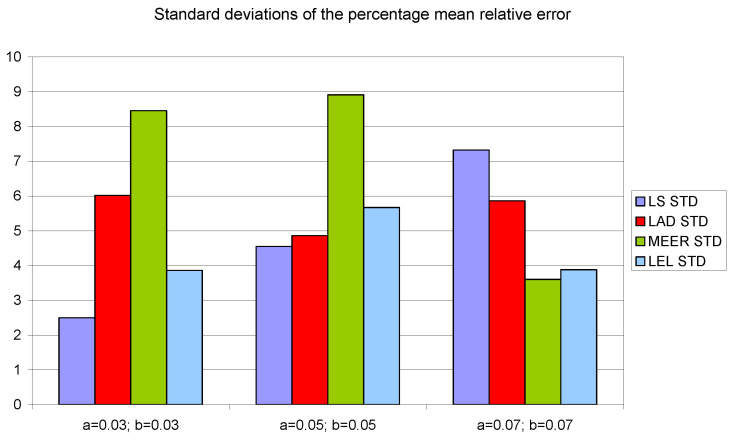
The standard deviations of the percentage mean relative error obtained as the average value of indicators for four different input signals using LS, LAD, MEER, and LEL estimators.

**Table 1 entropy-22-00834-t001:** The physical constraints and the plant model parameters.

Parameter	Value	Unit	Description
*h* _1,max_	4.00	[m]	Max. water level of tank 1
*h*_1,min_ = *h*_2,min_	0.00	[m]	Min. water level of tanks 1, 2
*h* _2,max_	2.00	[m]	Max. water level of tank 2
*h* _10_	0.75	[m]	Initial condition of tank 1
*h* _20_	0.50	[m]	Initial condition of tank 2
*a*_1_ = *a*_2_	0.05	[m]	Area of water outlet holes
*A* _1_	1.50	[m^2^]	Cross-section of tank 1
*A* _2_	0.75	[m^2^]	Cross-section of tank 2
*u* _0_	0.05	[m^3^/s]	Initial water inflow

**Table 2 entropy-22-00834-t002:** The mean value of the number of iterations for different input signals and estimators.

Mean Value of the Number of Iterations				
Estimator	LS	LAD	MEER	LEL
**step input**	25.92	37.06	17.85	27.17
**sinusoidal 0.25 Hz**	24.11	35.33	17.55	25.67
**sinusoidal 2.50 Hz**	24.40	34.68	17.53	26.86
**optimal input**	22.61	21.61	17.07	23.41

**Table 3 entropy-22-00834-t003:** The mean value of the number of function evaluations for different input signals and estimators.

Mean Value of the Number of Function Evaluations				
Estimator	LS	LAD	MEER	LEL
**step input**	97.13	178.31	60.67	100.63
**sinusoidal 0.25 Hz**	91.99	169.31	59.29	96.25
**sinusoidal 2.50 Hz**	92.72	168.88	58.63	87.37
**optimal input**	80.36	159.77	58.81	87.20

**Table 4 entropy-22-00834-t004:** The percentage mean relative error for different input signals and estimators.

Percentage Mean Relative Error				
Estimator	LS	LAD	MEER	LEL
**step input**	29.94%	22.96%	34.90%	33.88%
**sinusoidal 0.25 Hz**	30.03%	22.54%	26.62%	36.10%
**sinusoidal 2.50 Hz**	36.05%	21.49%	28.69%	36.66%
**optimal input**	25.48%	12.25%	13.74%	24.47%

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
