# Peer review of "Robustness Analysis of the Estimators for the Nonlinear System Identification"

_entropy, 2020, doi:10.3390/e22080834_

Round 1

Reviewer 1 Report

Article Review: Robustness analysis of Renyi-like estimator for non-linear system identification.

The paper Jakowluk and Golfewski investigates a very challenging problem of parameter inference of non-linear systems. The paper is split into two parts, in the first part the authors propose an extension to the Renyi entropy-based estimator, which they name to be Renyi-like, and in the second part, the authors design a example plant model (with fixed parameters) to demonstrate the effectiveness of theirs and other well-known estimators. Overall, the paper is written well, the authors describe their methodology really well and present their findings very objectively.

However, it is the reviewer’s suspicion that the authors sadly had a major oversight.   That is, the ‘novel’ estimator that they propose (minimum error entropy Renyi-like, MEERL (Eq. 19)) is in essence least squares in disguise. The authors come to the new estimate by interchanging the log and the summation in the classical minimum error entropy Renyi estimator (Eq 18). Then they choose their distribution to be normal centred at zero with variance one and evaluate their residual (Eq. 11) in this normal distribution. If one is to take the log of this normal distribution, then it is simple the residual squared. Hence, Eq. 19 is simply a scaled version of least squares. This is unfortunate, as it sadly discredits their estimator’s novelty.

The second part of the paper is the comparison of the different estimators. The authors do have a good setup; however, they need to reconsider their numerical experiments to give proper statistical insights. If the authors are to demonstrate that their conclusions are structural and not a function of the particular parameters which were chosen for the plant model, the authors need to consider repeating the experiment under multiple plant regimes. Then perform statistics on how the performance of the different methods change, and if there are any statistical differences between the approaches.

In conclusion, this is a good paper, however, the core story as the paper itself is not novel or robust. Hence, it is this reviewer’s opinion that the authors should restructure the paper and perform more numerical experiments.

Author Response

Reviewer #1
We would like to express sincere appreciation to the Reviewer for his/her insightful comments. We tried to do our best to make changes and improvements in the revised version of the manuscript, which meet the suggestions and requirements of the Reviewer.

Comments to the Author:
The paper Jakowluk and Golfewski investigates a very challenging problem of parameter inference of non-linear systems. The paper is split into two parts, in the first part the authors propose an extension to the Renyi entropy-based estimator, which they name to be Renyi-like, and in the second part, the authors design a example plant model (with fixed parameters) to demonstrate the effectiveness of theirs and other well-known estimators. Overall, the paper is written well, the authors describe their methodology really well and present their findings very objectively.
However, it is the reviewer’s suspicion that the authors sadly had a major oversight. That is, the ‘novel’ estimator that they propose (minimum error entropy Renyi-like, MEERL (Eq. 19)) is in essence least squares in disguise. The authors come to the new estimate by interchanging the log and the summation in the classical minimum error entropy Renyi estimator (Eq 18). Then they choose their distribution to be normal centred at zero with variance one and evaluate their residual (Eq. 11) in this normal distribution. If one is to take the log of this normal distribution, then it is simple the residual squared. Hence, Eq. 19 is simply a scaled version of least squares. This is unfortunate, as it sadly discredits their estimator’s novelty.

Thank you to the Reviewer for pointing this out. Of course one can easily prove that for continuous signals the results obtained based on the equation (19) should be similar to that obtained using equation (12). However, based on the numerical results presented in the Table 4 and Figure 11 this is not obvious to discrete signals.

The second part of the paper is the comparison of the different estimators. The authors do have a good setup; however, they need to reconsider their numerical experiments to give proper statistical insights. If the authors are to demonstrate that their conclusions are structural and not a function of the particular parameters which were chosen for the plant model, the authors need to consider repeating the experiment under multiple plant regimes. Then perform statistics on how the performance of the different methods change, and if there are any statistical differences between the approaches.

Thank you to the Reviewer. We have reconsidered numerical experiments to give a statistical insights. To demonstrate that the results are structural and are not a function of the particular model parameters the numerical experiments have been repeated for additional initial parameter values a = b = 0.03, and a = b= 0.07, respectively. To report the results of numerical experiments the bar graphs (Figures 9-11) have been used. A conclusions of these additional experiments are provided in the lines 346-354.

I would like to thank again the Reviewer for his/her insightful comments on the paper. I hope that the revised version of my manuscript is more comprehensive.

Reviewer 2 Report

Review of Paper 815691

"Robustness Analysis of Renyi-Like Estimator for Non-Linear System Identification"

by Wiktor Jakowluk and Karol Godlewski 

submitted to Entropy

This paper illustrates various estimation methods for the non-linear system identification under various excitation signals.

The proposed study is mainly based on numerical examples and on simulation results.

The topic treated in this paper is of interest to the readers of Entropy.

The paper is generally well written and clear.

However, some points need improvements:

- equation (10): a description of the function f and its properties should be provided;

- equation (13): the model parameter estimation based on Least Absolute Deviations should be based on the minimization of the sum of absolute values of the residuals; it seems that the absolute value is missing;

- equations (18) and (19): specify the range of parameter alpha;

- lines 237 and 249: reference is made to Figure 1, whereas I guess the proper one is Figure 4.

Author Response

Reviewer #2
I would like to express sincere appreciation to the Reviewer for his/her insightful comments. I tried to do my best to make changes and improvements in the revised version of the manuscript, which meet the suggestions and requirements of the Reviewer.

Comments to the Author:
This paper illustrates various estimation methods for the non-linear system identification under various excitation signals. The proposed study is mainly based on numerical examples and on simulation results. The topic treated in this paper is of interest to the readers of Entropy. The paper is generally well written and clear.
However, some points need improvements: equation (10): a description of the function f and its properties should be provided;

Thank you to the Reviewer. A description of the function f is given in lines 190-191.

equation (13): the model parameter estimation based on Least Absolute Deviations should be based on the minimization of the sum of absolute values of the residuals; it seems that the absolute value is missing;

Thank you to the Reviewer. This was an error and it has been corrected, line 198.

equations (18) and (19): specify the range of parameter alpha;

Thank you to the Reviewer. The range of parameter alpha has been specified in lines 219, and 231.

lines 237 and 249: reference is made to Figure 1, whereas I guess the proper one is Figure 4.

Thank you to the Reviewer. The system parameter identification block diagram is shown in Figure 1. I reformulated the sentences to make them more clearer, lines 248, and 260.

I would like to thank the Reviewer for his/her insightful comments on the paper. I hope that the revised version of our manuscript is more precise.

Reviewer 3 Report

The Introduction must be improved: 

1) The system that is studied is not motivated and no references are given about this system or previous investigations of similar system.

2) Refernces to previous work that compares different estiamators should be included; if there are no such work in the literature it should be stated.

 The description of the research design can be improved: It is difficult to judge since the studied system and the used system parameters are not motivated.

The presentation of the results could be more clearly presented:  Are there any other comparative studies of different estiamators or any other work on the studied system (the tanks) that one could compare with?

Somde details: 

line 66: "... as precisely as possible..." define precisely. 

line 87: MEERL used on line 87, before it was defined on line 89. 

line 105: "...constructing an exact mathematical model..." Can an exact model be constructed;pls rephrase. 

line 129: give reference for this model of the water tank (applies to the entire section and all the equations) 

Table 1. Motivate why these paramter values were used. 

Section 5: again, why were these parameter values used? 

Table 4: here 2 significant digits are given. Above (table 2 nd 3) 4 significant digits were given. Be consistent and use reasonable number of significant digits. 

line 315: "experimental results". No experiemtns were described. explain or rephrase

line 345: "It has been proven that the sinusoidal excitation signals are not recommended for dynamic system identification in the time domain." it is a very broad statement. I suggest to be more specific. 

Author Response

Reviewer #3
We would like to express sincere appreciation to the Reviewer for his/her insightful comments. We tried to do our best to make changes and improvements in the revised version of the manuscript, which meet the suggestions and requirements of the Reviewer.

Comments to the Author:
The Introduction must be improved:
1) The system that is studied is not motivated and no references are given about this system or previous investigations of similar system.

Thank you to the Reviewer. The studied system has been motivated and the references for the identical, and similar systems have been attached in lines 88-99.

2) Refernces to previous work that compares different estiamators should be included; if there are no such work in the literature it should be stated.
Thank you to the Reviewer for this question. Similar publications on non-linear double tank system including different estimators comparison were not found. It was stated in the article.

The description of the research design can be improved: It is difficult to judge since the studied system and the used system parameters are not motivated. The presentation of the results could be more clearly presented: Are there any other comparative studies of different estiamators or any other work on the studied system (the tanks) that one could compare with?

We have improved the Introduction and the Numerical Results and Discussion parts to be more understandable. The publications regarding water tanks systems have been included [27-29].

Some details:
ï‚· line 66: "... as precisely as possible..." define precisely.
Thank you to the Reviewer. It has been reformulated in line 65.

ï‚· line 87: MEERL used on line 87, before it was defined on line 89.
Thank you to the Reviewer. It has been corrected in line 85.

ï‚· line 105: "...constructing an exact mathematical model..." Can an exact model be constructed; pls rephrase.
Thank you to the Reviewer. It has been rephrased in line 112.

ï‚· line 129: give reference for this model of the water tank (applies to the entire section and all the equations).
Thank you to the Reviewer. The link to mathworks side has been added [29].

ï‚· Table 1. Motivate why these paramter values were used.
Thank you to the Reviewer. It is a model of a laboratory system and the parameters are selected arbitrarily to ensure the gravitational flow of water through the tanks.

ï‚· Section 5: again, why were these parameter values used?
Thank you to the Reviewer. For other model parameters, the course of the liquid flow phenomenon will be the same.

ï‚· Table 4: here 2 significant digits are given. Above (table 2 nd 3) 4 significant digits were given. Be consistent and use reasonable number of significant digits.
Thank you to the Reviewer. It has been corrected.

ï‚· line 315: "experimental results". No experiemtns were described. explain or rephrase.
Thank you to the Reviewer. It has been rephrased.

ï‚· line 345: "It has been proven that the sinusoidal excitation signals are not recommended for dynamic system identification in the time domain." it is a very broad statement. I suggest to be more specific.
Thank you to the Reviewer. The sentence has been written in a different way.

Thank you again to the Reviewer and I hope that the revised version of my manuscript is more precise.

Round 2

Reviewer 1 Report

Article Second Review: Robustness analysis of Renyi-like estimator for non-linear system identification.

Recapping from the previous review, the paper Jakowluk and Golfewski investigate a very challenging problem of parameter inference of non-linear systems. The paper is split into two parts, in the first part the authors propose an extension to the Renyi entropy-based estimator, which they name to be Renyi-like, and in the second part, the authors design a example plant model (with fixed parameters) to demonstrate the effectiveness of theirs and other well-known estimators. Overall, the paper is written well, the authors describe their methodology really well and present their findings very objectively.

This reviewer raised two major concerns in the paper, the first was the novelty form of the MEERL method, and the second being the lack of repeated experimental scenarios to substantiate their claim.  In the revised version of the manuscript, the authors sadly did not address the first point appropriately, however, they did change their experimental design and presented new results which help the author’s claims.  We will now breakdown the two points in more detail.

Regarding the novelty of the MEERL, the reviewers reply is not correct. The issue is not the form of the MEERL or the data which it evaluates. The issue is that mathematically, the condition used in this paper (MEERL with alpha = 2.0 and p is Gaussian) is proportional to Least Squares (LS). This is the point that the reviewers did not address. Hence, MEERL in this paper, in these experiments is proportional to LS.  To argue against this, the authors need to show that Equation 19 and Equation 12 are not proportional to each other under the assumptions made in this paper.

Regarding the repeated experiment, the authors changed the setup of the double tank system and performed multiple run to gain average estimates of key readouts such as number of iterations, functional evaluations, and relative error. The authors should also include error bars (+- STD) to demonstrate the significance of the different approaches.

It is this reviewer’s opinion that this paper be only accepted as an investigation of different estimators for the non-linear double tank system. This is a well written paper, and the analysis they do is a contribution to the community. However, their proposal for MEERL is not substantiated. Hence, it is this reviewer’s recommendation for minor revisions, with the removal the MEERL from the paper, and replace it with MEER (the results are probably already generated by the authors), and the paper stand on its strength: “Robustness Analysis of the Estimators for the non-linear double water tank system”.  

Author Response

We would like to express sincere appreciation to the Reviewer for his/her insightful comments. We tried to do our best to make changes and improvements in the revised version of the manuscript, which should meet the suggestions and requirements of the Reviewer.

Article Second Review: Robustness analysis of Renyi-like estimator for non-linear system identification.

Comments to the Author:

Recapping from the previous review, the paper Jakowluk and Golfewski investigate a very challenging problem of parameter inference of non-linear systems. The paper is split into two parts, in the first part the authors propose an extension to the Renyi entropy-based estimator, which they name to be Renyi-like, and in the second part, the authors design a example plant model (with fixed parameters) to demonstrate the effectiveness of theirs and other well-known estimators. Overall, the paper is written well, the authors describe their methodology really well and present their findings very objectively.

This reviewer raised two major concerns in the paper, the first was the novelty form of the MEERL method, and the second being the lack of repeated experimental scenarios to substantiate their claim.  In the revised version of the manuscript, the authors sadly did not address the first point appropriately, however, they did change their experimental design and presented new results which help the author’s claims.  We will now breakdown the two points in more detail.

Regarding the novelty of the MEERL, the reviewers reply is not correct. The issue is not the form of the MEERL or the data which it evaluates. The issue is that mathematically, the condition used in this paper (MEERL with alpha = 2.0 and p is Gaussian) is proportional to Least Squares (LS). This is the point that the reviewers did not address. Hence, MEERL in this paper, in these experiments is proportional to LS.  To argue against this, the authors need to show that Equation 19 and Equation 12 are not proportional to each other under the assumptions made in this paper.

Thank you to the Reviewer. I am sorry, but I am unable to prove that equations 12 and 19 are not proportional to each other for discrete signals. Obviously, one can easily prove that these equations are proportional to continuous signals (under the assumptions: MEERL with alpha = 2.0 and p is Gaussian). Therefore, we have decided to remove an equation 19, and figure 4 with comments. We have replaced the MEERL estimator data with the classic MEER estimator data, and we have made additional computations to enable the comparison of the MEER estimator with the others.

Regarding the repeated experiment, the authors changed the setup of the double tank system and performed multiple run to gain average estimates of key readouts such as number of iterations, functional evaluations, and relative error. The authors should also include error bars (+- STD) to demonstrate the significance of the different approaches.

Thank you to the Reviewer. Using new and previous computational results, we have calculated the standard deviations and plotted them utilizing bar charts. Some conclusions about the STD have been made, and then have been highlighted in red color.

It is this reviewer’s opinion that this paper be only accepted as an investigation of different estimators for the non-linear double tank system. This is a well written paper, and the analysis they do is a contribution to the community. However, their proposal for MEERL is not substantiated. Hence, it is this reviewer’s recommendation for minor revisions, with the removal the MEERL from the paper, and replace it with MEER (the results are probably already generated by the authors), and the paper stand on its strength: “Robustness Analysis of the Estimators for the non-linear double water tank system”.  

Thank you to the Reviewer for the suggestions for improving the manuscript. Concluding, we have revised the title of the article according to the wishes of the Reviewer. We have removed an equation (19), and figure 4 with comments (i.e. the part related to the MEERL estimator). We have performed additional calculations for the MEER estimator, and then we have changed data in tables 2-4. We have also modified charts shown in figures 6-7, and 8-10. We have plotted new STD charts 11-13 according to the reviewer suggestion. Finally, some comments have been made subject to new calculation results. All, new results and comments have been highlighted in red color.

I would like to thank again the Reviewer for his/her insightful comments on the paper. I hope that the revised version of my manuscript meets the expectations of the Reviewer.

Reviewer 2 Report

The corrections have been performed in a satisfactorily manner. 

In my opinion the paper can be accepted for publication

Author Response

Comments and Suggestions for Authors:

The corrections have been performed in a satisfactorily manner. 

In my opinion the paper can be accepted for publication.

We would like to express sincere appreciation to the Reviewer for his/her insightful comments. We are grateful that in its current version the the manuscript meets the Reviewer's requirements.
